# Monitoring Spatial and Temporal Variabilities of Gross Primary Production Using MAIAC MODIS Data

**Marcos Fernández-Martínez [1,*] , Rong Yu [2], John Gamon [2,3], Gabriel Hmimina [2], Iolanda Filella [4,5], Manuela Balzarolo [4,5], Benjamin Stocker [4,5] and Josep Peñuelas [4,5]**

1. Centre of Excellence PLECO (Plants and Ecosystems), Department of Biology, University of Antwerp, 2610 Wilrijk, Belgium
2. Center for Advanced Land Management Information Technologies, School of Natural Resources, University of Nebraska-Lincoln, Lincoln, NE 68583, USA; rong.yu@unl.edu (R.Y.); jgamon@unl.edu (J.G.); hmimina@unl.edu (G.H.)
3. Earth & Atmospheric Sciences, and Biological Sciences, University of Alberta, Edmonton, AB T6H-0Z3, Canada
4. CSIC, Global Ecology Unit, CREAF-CSIC-UAB, Cerdanyola del Vallès, 08193 Barcelona, Catalonia, Spain; iola@creaf.uab.cat (I.F.); Manuela.Balzarolo@uantwerpen.be (M.B.); benjamin.stocker@gmail.com (B.S.); Josep.Penuelas@uab.cat (J.P.)
5. CREAF, Cerdanyola del Vallès, 08193 Barcelona, Catalonia, Spain
* Correspondence: marcos.fernandez-martinez@uantwerpen.be

**Abstract:** Remotely sensed vegetation indices (RSVIs) can be used to efficiently estimate terrestrial primary productivity across space and time. Terrestrial productivity, however, has many facets (e.g., spatial and temporal variability, including seasonality, interannual variability, and trends), and different vegetation indices may not be equally good at predicting them. Their accuracy in monitoring productivity has been mostly tested in single-ecosystem studies, but their performance in different ecosystems distributed over large areas still needs to be fully explored. To fill this gap, we identified the facets of terrestrial gross primary production (GPP) that could be monitored using RSVIs. We compared the temporal and spatial patterns of four vegetation indices (NDVI, EVI, NIR$_V$, and CCI), derived from the MODIS MAIAC data set and of GPP derived from data from 58 eddy-flux towers in eight ecosystems with different plant functional types (evergreen needle-leaved forest, evergreen broad-leaved forest, deciduous broad-leaved forest, mixed forest, open shrubland, grassland, cropland, and wetland) distributed throughout Europe, covering Mediterranean, temperate, and boreal regions. The RSVIs monitored temporal variability well in most of the ecosystem types, with grasslands and evergreen broad-leaved forests most strongly and weakly correlated with weekly and monthly RSVI data, respectively. The performance of the RSVIs monitoring temporal variability decreased sharply, however, when the seasonal component of the time series was removed, suggesting that the seasonal cycles of both the GPP and RSVI time series were the dominant drivers of their relationships. Removing winter values from the analyses did not affect the results. NDVI and CCI identified the spatial variability of average annual GPP, and all RSVIs identified GPP seasonality well. The RSVI estimates, however, could not estimate the interannual variability of GPP across sites or monitor the trends of GPP. Overall, our results indicate that RSVIs are suitable to track different facets of GPP variability at the local scale, therefore they are reliable sources of GPP monitoring at larger geographical scales.

**Keywords:** GPP; seasonality; interannual variability; trends; forests

## 1. Introduction

The continuous development of remote-sensing techniques has increased their popularity in the fields of earth science, ecology, conservation, and nature and land management [1–8]. Remotely sensed vegetation indices (RSVIs) are frequently used to broadly and efficiently monitor spatial and temporal variations in terrestrial primary productivity (e.g., ecosystem photosynthesis). The accuracies of the RSVIs, however, have only been evaluated using few ecosystems using one or two indices [9–12]. Broad-scale and global studies often cannot accurately assess the relationships between productivity and vegetation indices [1,13–15], so how well these relationships between vegetation indices and productivity can be transferred to different ecosystems is still not clear.

Terrestrial primary productivity (in a broad sense) has many facets and sources of variability, and RSVIs may not be equally good at monitoring all of them. Several studies have reported good relationships between the spatial variability of biomass stocks or the production of aboveground biomass and RSVIs such as the normalised difference vegetation index (NDVI) and the enhanced vegetation index (EVI) for boreal forests [16], shrublands [17], grasslands [10,18,19], and croplands [20]. Interannual variability of tree growth and fruit production have also been satisfactorily monitored by various RSVIs in Mediterranean evergreen broadleaved forests [2,12], but few studies have compared the performances of RSVIs across multiple ecosystems.

Other sources of variability of terrestrial productivity have been studied less. Little effort has been devoted to testing the relationship between the spatial variability of RSVI seasonality and the seasonality of terrestrial photosynthesis, despite the extensive focus on the study of phenology using RSVIs [6,9,21]. The performance of RSVIs across sites with different seasonalities therefore requires further study, comparing their performances using raw values and deseasonalised time series (anomalies). Several factors can disrupt this relationship and require further investigation, even though trend analyses using RSVIs are relatively common and have often identified a positive relationship with actual greenness or increase in biomass [1,15].

The choice of RSVI to use in a study is not always straightforward, because all RSVIs have both advantages and disadvantages. NDVI and EVI are currently two of the most widely used vegetation indices. NDVI is useful for comparing seasonal and interannual changes in vegetation growth and activity [12,19]. NDVI, however, can saturate in conditions of large biomass [22], a problem that EVI, which was designed to improve sensitivity in regions with large biomass, overcomes [23]. Recent studies of RSVIs have provided two new indices with very promising properties: the near-infrared reflectance index, $NIR_v$, [24], and the chlorophyll-carotenoid index, CCI [9]. $NIR_v$ addresses the confounding effects of leaf area, background brightness, and the distribution of photosynthetic capacity with depth in canopies and has recently been demonstrated to correlate well with gross primary production (GPP) across biomes [24]. CCI, which is based on changes in the content of chlorophyll and carotenoid pigments in leaves, can effectively monitor photosynthetic phenology in evergreen conifers, despite their small changes in crown cover [9].

We identified the features of terrestrial GPP that can be reliably monitored using RSVIs, comparing the temporal patterns of the indices and GPP derived from 58 eddy-flux towers distributed throughout Europe. We used four vegetation indices (NDVI, EVI, $NIR_v$, and CCI) calculated on weekly, monthly, and annual bases and tested whether they were able to monitor: (i) weekly and monthly temporal variability, using raw and deseasonalised time series of GPP and RSVIs, (ii) spatial variability, calculating the average annual value for each site, (iii) site seasonality, by calculating the average monthly intra-annual variability for each site, (iv) interannual variability, as the average variability amongst years for each site, and (v) trends in annual means for each site. We focused on identifying the differences amongst the performances of the RSVIs to estimate various facets of GPP across the vegetation types. Our results will therefore help researchers to select the best RSVI to use in each case and to justify their use for a particular purpose. Our results could also be used to improve GPP models by identifying the best method to parameterise these models with RSVIs.

## 2. Materials and Methods

### 2.1. Carbon Flux, Weather, and Remotely Sensed Data

We downloaded daily data for 58 European sites from the FLUXNET 2015 (http://fluxnet.fluxdata.org/data/download-data) Tier 1 data set containing data for weather and $CO_2$ flux from the eddy-covariance towers. The 58 sites (Figure 1, Table S1) covered boreal (six sites), temperate (43), and Mediterranean (nine) biomes and eight vegetation types (evergreen needle-leaved forest (16 sites), evergreen broad-leaved forest (three), deciduous broad-leaved forest (ten), mixed forest (three), open shrubland (two), grassland (nine), cropland (ten), and wetland (five)). We used all daily values of GPP from methods of daytime partitioning following [24]. We then calculated weekly, monthly, and annual values as the sum of GPP for each period (Figure 2). We also calculated average meteorological parameters for each site (mean temperature, total precipitation, and mean vapour-pressure deficit (VPD)) to use as predictors in statistical models (see descriptions below).

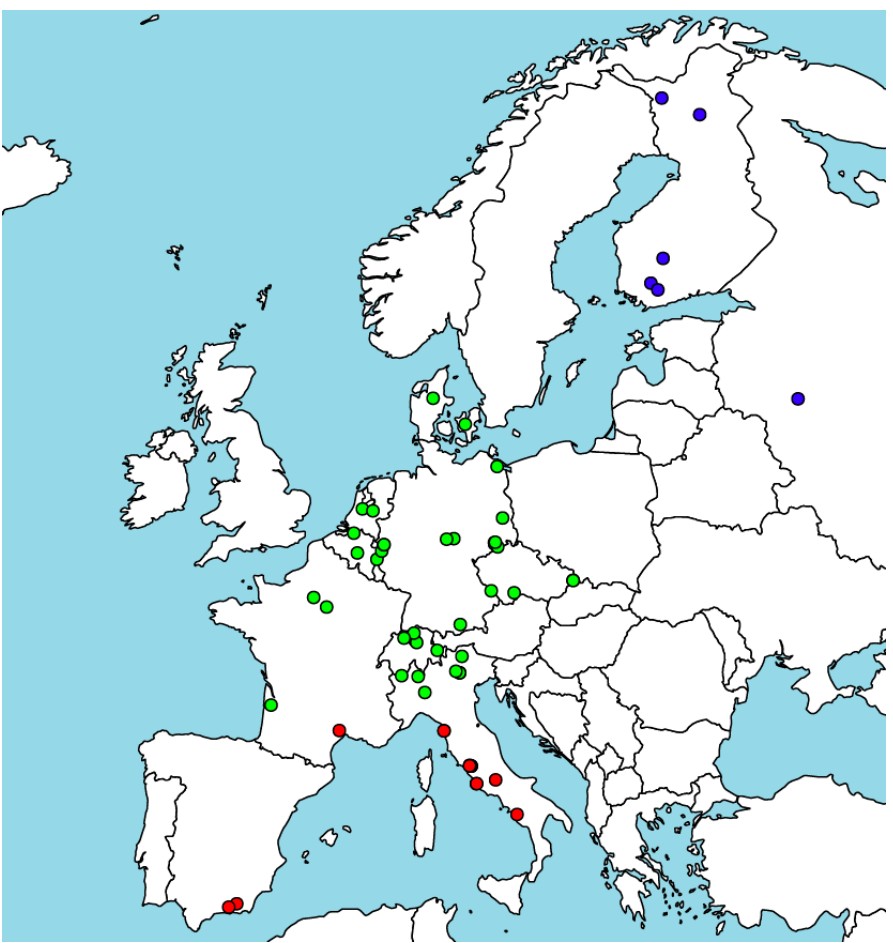

**Figure 1.** Map of the sites. Blue, green, and red dots correspond to boreal, temperate, and Mediterranean sites, respectively.

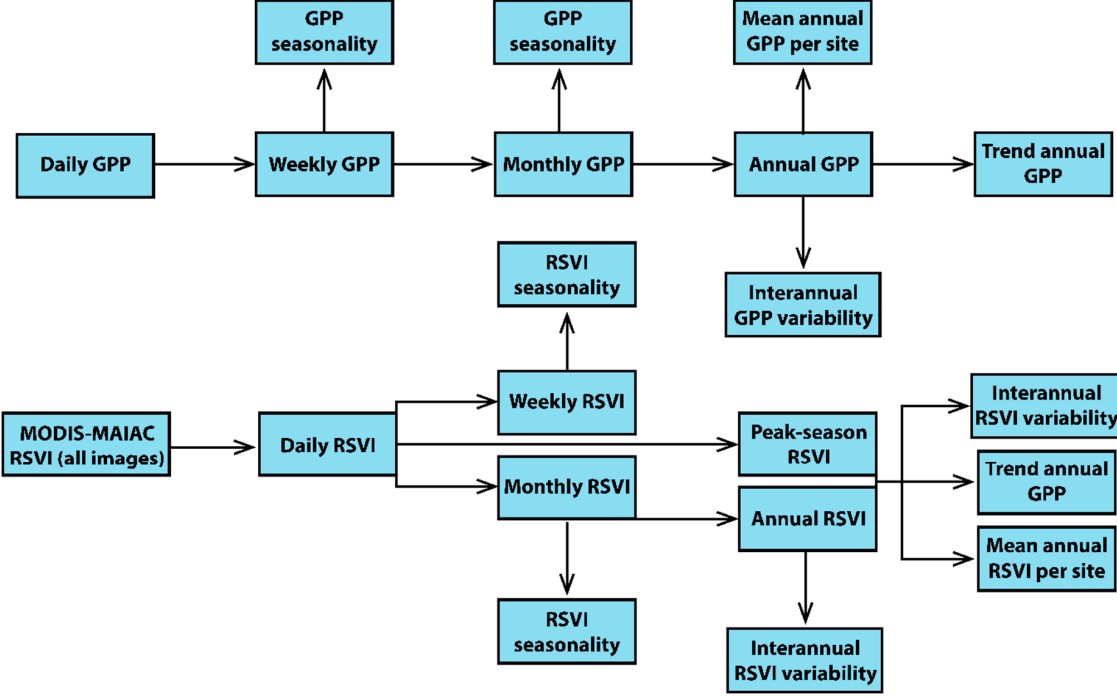

**Figure 2.** Flowchart indicating the steps followed to perform the analyses presented in this study. *Acronyms*: Gross primary production (GPP), remotely sensed vegetation indices (RSVI).

We downloaded all images for Europe for 2000–2016 from a MODIS (Moderate Resolution Imaging Spectroradiometer) composite processed by the MAIAC (Multi-Angle Implementation of Atmospheric Correction) [25–27] data set with a resolution of 1 km. The coarse resolution of the MAIAC dataset is not a problem when monitoring data coming from eddy covariance towers because land-use area around the towers is normally homogeneous [28]. We used raw data from the MAIAC data set, because this data set has less atmospheric noise and detects clouds better than previous data sets [25–27]. We therefore did not further filter our data. We calculated NDVI, EVI [23], $NIR_v$ as defined by [24] and [29], and CCI [9] using the central pixel from all records from the flux towers (Aqua + Terra images) by combining reflectance bands as:

$$NDVI = \frac{B2 - B1}{B2 + B1} \tag{1}$$

$$EVI = \frac{2.5 * (B2 * 0.0001 - B1 * 0.0001)}{B2 * 0.0001 + 6 * B1 * 0.0001 - 7.5 * B3 * 0.0001 + 1} \tag{2}$$

$$NIR_v = (NDVI - 0.08) * B2 \tag{3}$$

$$CCI = \frac{B11 - B1}{B11 + B1} \tag{4}$$

where the bandwidths for bands B1, B2, B3, B4, and B11 are 620–670, 841–876, 459–479, 545–565, and 526–536 nm, respectively. We calculated the RSVIs on weekly and monthly bases by first calculating the averages per day (more than one satellite overpass was often recorded, but calculating the median would have been nonsensical for two or three images per day) and then calculating the median of each index per week and month (Figure 2). We did not use maximum or mean values to estimate weekly and monthly values to avoid the interference of outliers. We also used weekly and monthly time resolutions instead of a daily resolution to prevent outliers and artefacts in the data. The CCI values were adjusted by (1 + CCI)/2, and $NIR_v$ was divided by its maximum value, for comparing to the NDVI and EVI values (which range from 0 to 1).

## 2.2. Statistical Analysis

The performance of the RSVIs for the predicting weekly and monthly variabilities of GPP was tested using GPP and RSVI time series, and linear regressions were fitted for each site and index, with GPP as the response variable and the RSVIs as the predictors. Only time series with 60 or more time-steps were considered for both the monthly and weekly analyses. We extracted the variance explained ($R^2$) from each regression, so we had four models for each site (testing the four RSVIs). We deseasonalised the weekly (monthly) time series of GPP and RSVI by subtracting the average and dividing by the standard deviation for each week (month) and site for each observation (i.e., we calculated the standardised anomalies of the time series) to determine if the relationship between weekly (monthly) GPP and RSVI was preserved when the annual seasonality of the time series was removed. We then performed the same analysis using the standardised anomalies to produce eight additional measures of goodness of fit (i.e., 16 per site in total). We tested the potential effect of snow cover on our results by repeating these analyses using times series without winter values (i.e., weeks 1–12 and 46–52 were removed for the weekly analyses, and December, January, February, and March were removed for the monthly analyses). Weeks were calculated from 1 January.

Our results would suggest that the RSVIs were able to track the temporal variability of GPP independently of their seasonal cycle if the models using the standardised anomalies performed as well as those using data with the seasonal component. If the performance using anomalies was dramatically lower than the performance using data with the seasonal component, our results would identify the RSVIs that could track the seasonal component of GPP well, but not its seasonal anomalies (deviations from the mean).

We further identified the factors that determined the goodness of fit of the relationship between GPP and RSVIs for each site, using data both with and without the seasonal component (i.e., standardised anomalies). We first calculated several potentially influencing variables for each site such as climatic variables (mean annual temperature (MAT), mean annual precipitation (MAP), mean annual VPD), the seasonalities and interannual variabilities of GPP and the RSVIs, and the variances of GPP and the RSVIs explained by the seasonal cycle (calculated from a linear model such as GPP or RSVI as a function of month (as a factor of 12 levels)). Seasonality (intra-annual variability) was calculated as the proportional variability (PV) index [30,31] of the averages of a variable for the twelve months of the year. PV is an index of variability, ranging from 0 (no variability) to 1 (very high variability), that assesses the proportional variability of a data set by calculating all pair-wise comparisons between the values of a variable. This index overcomes some of the mathematical issues of the coefficient of variation [30]. The interannual variabilities of GPP and the RSVIs were calculated for each site as the average of the interannual PV for each month. We then identified the factors that were correlated with the goodness of fit of the relationship between GPP and an RSVI (with and without seasonality) using linear models (e.g., NDVI-GPP performance for each site as a function of site GPP seasonality + NDVI seasonality + MAT + MAP + VPD).

We calculated (i) the average annual GPP, (ii) the average annual values of the RSVIs across months, and (iii) the average peak-season values of the RSVIs for each site to determine how well the RSVIs explained the spatial variability of GPP. We only used years with data for all 12 months in which both GPP and the RSVIs were recorded. We compared the results using the averages and the peak-season values of the RSVIs to identify artefacts associated with snow, standing water, standing dead vegetation, and other potential confounding factors. Only sites with five or more years of data were used to calculate the annual averages. Generalised Additive Models (GAM) were used for estimating the relationships between GPP and the RSVIs to test for nonlinear responses.

We then determined if the seasonalities of the RSVIs (average intra-annual variability) were correlated with the seasonality of GPP using linear models. Seasonality was estimated as the PV of the 12 monthly values within a year, averaged over all complete years in which GPP and RSVI data were available. We next determined if the interannual variability of GPP was correlated with the interannual variability of the RSVIs, also using linear models. The interannual variabilities of GPP and the RSVIs

were calculated as the interannual monthly PV of each month averaged over the 12 months of a year. For both seasonality and interannual variability, we only used sites in which monthly time series of GPP and the RSVIs were available for five or more years [31]. Finally, we determined if the trends in annual GPP could be inferred from the trends in the annual average and the annual peak-season values of the RSVIs by calculating the annual trends of GPP and the annual average and peak-season RSVIs of the sites for which the time series were at least seven years [32]. Trends were calculated using the robust Theil-Sen's slope estimator [33].

All statistical analyses were performed with R statistical software [34]. The amount of variance explained by the predictors in a model was assessed using the "*lmg*" metric [35] in the "*relaimpo*" package [36].

## 3. Results

### 3.1. Monitoring Weekly and Monthly Temporal Variability of GPP Using the RSVIs

All RSVIs correlated well with the weekly and monthly time series of GPP, but the correlations were slightly higher using the monthly than the weekly data (Figure 3, Table 1). The performances between the ecosystem types did not differ greatly, except the correlations for both weekly and monthly data were consistently highest and lowest for grasslands and evergreen broad-leaved forests, respectively. EVI and $NIR_v$ monitored the variability of GPP the best, accounting for $\geq$48% of the variance. CCI was the index, explaining the lowest variance in GPP for all ecosystem types using either weekly or monthly data. CCI, nonetheless, still accounted for averages of 19 and 28% of the GPP variability for the weekly and monthly time series, respectively. Only $\leq$15% of the variability of GPP, however, could be explained using the RSVIs when the same analysis was repeated after removing the seasonal component of the GPP and RSVI time series across all ecosystem types. Most of the ecosystem types, though, had no correlations. All RSVIs performed very similarly: the correlations between GPP and the RSVIs nearly disappeared when weekly or monthly anomalies were used. These results remained consistent after winter values from the analyses were removed to avoid biases because of snow cover (Table S2).

The seasonal component of the time series, however, explained a large amount of the variance of monthly GPP (mean $\pm$ standard error, 79.0 $\pm$ 1.7%) and of the RSVIs (NDVI: 74.6 $\pm$ 1.7%, EVI: 82.2 $\pm$ 1.6%, $NIR_v$: 84.3 $\pm$ 1.2%, CCI: 70.0 $\pm$ 1.7%). Again, the results did not differ substantially for the weekly or monthly data. The RSVIs, therefore, monitored the seasonal cycle of GPP very well, at both weekly and monthly scales, but largely failed to monitor the anomalies of the GPP time series. Our analyses found that the performance of the RSVIs in explaining the variability of GPP, including its seasonal component, was better at sites where more of the variances of GPP and the RSVIs were explained by their seasonal component (Table 2). The interannual variability of GPP was also positively correlated with the goodness of fit of its relationships with the RSVIs using both monthly and weekly time series. A higher seasonality of GPP was also positively correlated with a better performance of NDVI and CCI in the weekly time series. CCI performed better at sites with larger GPP fluxes, and NDVI performed the best for grasslands. The interannual variability of GPP was the main factor explaining the performances of EVI, $NIR_v$, and CCI when the deseasonalised weekly time series was used. Only $NIR_v$ retained the same relationship when the monthly deseasonalised time series was used, and the interannual variability of EVI was positively correlated with the EVI performance when the weekly and monthly data were used. Climate had a very low influence on the relationship between RSVIs and GPP. Vapour-pressure deficit was positively linked to better performance of EVI and CCI using deseasonalised weekly time series. The same result was found for mean annual temperature when using deseasonalised monthly EVI time series.

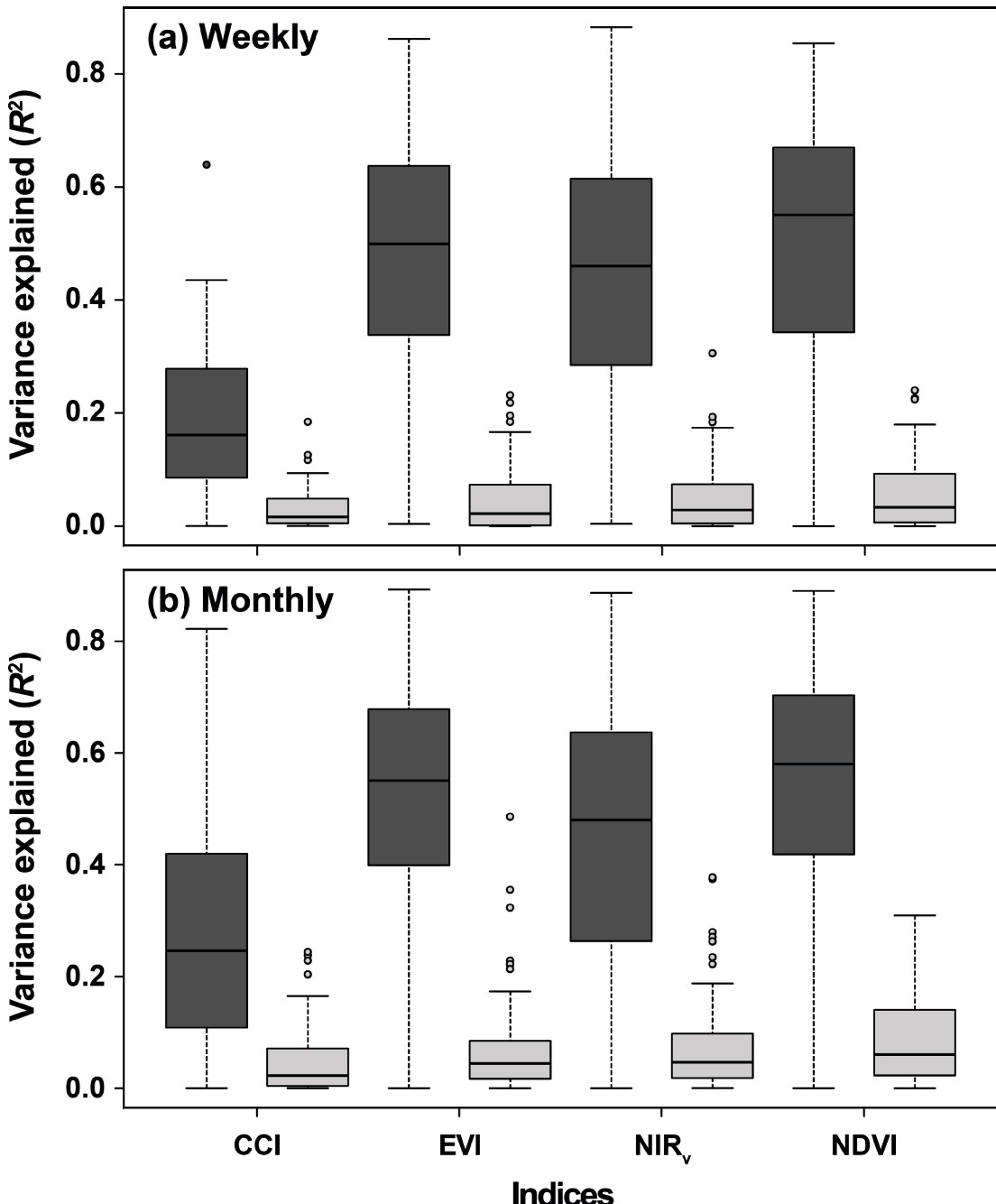

**Figure 3.** Boxplots of the variance explained ($R^2$) by the vegetation indices in the weekly (**a**) and monthly (**b**) raw (dark grey) and deseasonalised (light grey) time series of gross primary production (GPP).

**Table 1.** Amounts of variance explained ($R^2$, mean $\pm$ standard error of the mean) by NDVI, EVI, NIR$_v$, and CCI for the raw and deseasonalised weekly and monthly time series of GPP. Significant coefficients are highlighted in bold. ENF, evergreen needleleaved forest; EBF, evergreen broadleaved forest; DBF, deciduous broadleaved forest; MF, mixed forest; OSH, open shrubland; GRA, grassland; CRO, cropland; WET, permanent wetland.

| Ecosystem | NDVI | EVI | NIR$_v$ | CCI | N |
|---|---|---|---|---|---|
| *Weekly* | | | | | |
| ENF | **0.42 $\pm$ 0.07** | **0.43 $\pm$ 0.05** | **0.46 $\pm$ 0.06** | **0.13 $\pm$ 0.03** | 16 |
| EBF | **0.19 $\pm$ 0.07** | **0.20 $\pm$ 0.08** | 0.19 $\pm$ 0.08 | **0.11 $\pm$ 0.02** | 3 |
| DBF | **0.42 $\pm$ 0.09** | **0.53 $\pm$ 0.10** | **0.54 $\pm$ 0.10** | **0.24 $\pm$ 0.07** | 10 |
| MF | **0.48 $\pm$ 0.01** | **0.68 $\pm$ 0.01** | **0.66 $\pm$ 0.01** | **0.30 $\pm$ 0.07** | 3 |
| OSH | 0.51 $\pm$ NA | 0.47 $\pm$ NA | 0.50 $\pm$ NA | 0.00 $\pm$ NA | 2 |
| GRA | **0.62 $\pm$ 0.05** | **0.56 $\pm$ 0.05** | **0.55 $\pm$ 0.05** | **0.25 $\pm$ 0.04** | 9 |
| CRO | **0.35 $\pm$ 0.06** | **0.39 $\pm$ 0.05** | **0.40 $\pm$ 0.06** | **0.14 $\pm$ 0.03** | 10 |
| WET | **0.48 $\pm$ 0.07** | **0.55 $\pm$ 0.07** | **0.56 $\pm$ 0.08** | **0.25 $\pm$ 0.05** | 5 |
| Mean | **0.43 $\pm$ 0.03** | **0.48 $\pm$ 0.03** | **0.49 $\pm$ 0.03** | **0.19 $\pm$ 0.02** | 58 |
| *Weekly anomalies* | | | | | |
| ENF | **0.03 $\pm$ 0.01** | 0.02 $\pm$ 0.01 | **0.03 $\pm$ 0.01** | 0.01 $\pm$ 0.00 | 16 |
| EBF | 0.02 $\pm$ 0.01 | 0.01 $\pm$ 0.01 | 0.01 $\pm$ 0.01 | 0.06 $\pm$ 0.03 | 3 |
| DBF | **0.07 $\pm$ 0.02** | **0.07 $\pm$ 0.02** | **0.07 $\pm$ 0.02** | **0.07 $\pm$ 0.01** | 10 |
| MF | 0.02 $\pm$ 0.01 | 0.01 $\pm$ 0.01 | 0.02 $\pm$ 0.02 | 0.01 $\pm$ 0.00 | 3 |
| OSH | 0.05 $\pm$ NA | 0.05 $\pm$ NA | 0.03 $\pm$ NA | 0.00 $\pm$ NA | 2 |
| GRA | **0.10 $\pm$ 0.03** | **0.08 $\pm$ 0.02** | **0.09 $\pm$ 0.03** | 0.02 $\pm$ 0.01 | 9 |
| CRO | **0.08 $\pm$ 0.02** | **0.11 $\pm$ 0.03** | **0.10 $\pm$ 0.03** | **0.06 $\pm$ 0.02** | 10 |
| WET | 0.00 $\pm$ 0.00 | 0.00 $\pm$ 0.00 | 0.01 $\pm$ 0.00 | 0.02 $\pm$ 0.01 | 5 |
| Mean | **0.05 $\pm$ 0.00** | **0.05 $\pm$ 0.01** | **0.06 $\pm$ 0.01** | **0.03 $\pm$ 0.01** | 58 |
| *Monthly* | | | | | |
| ENF | **0.43 $\pm$ 0.06** | **0.49 $\pm$ 0.06** | **0.50 $\pm$ 0.06** | **0.20 $\pm$ 0.04** | 16 |
| EBF | 0.16 $\pm$ 0.07 | 0.23 $\pm$ 0.12 | 0.23 $\pm$ 0.12 | **0.10 $\pm$ 0.04** | 3 |
| DBF | **0.43 $\pm$ 0.09** | **0.55 $\pm$ 0.11** | **0.56 $\pm$ 0.11** | **0.35 $\pm$ 0.10** | 10 |
| MF | **0.45 $\pm$ 0.03** | **0.77 $\pm$ 0.02** | **0.74 $\pm$ 0.02** | **0.48 $\pm$ 0.06** | 3 |
| OSH | 0.32 $\pm$ NA | 0.27 $\pm$ NA | 0.35 $\pm$ NA | 0.21 $\pm$ NA | 2 |
| GRA | **0.61 $\pm$ 0.06** | **0.65 $\pm$ 0.04** | **0.65 $\pm$ 0.04** | **0.42 $\pm$ 0.07** | 9 |
| CRO | **0.41 $\pm$ 0.07** | **0.38 $\pm$ 0.06** | **0.43 $\pm$ 0.06** | **0.21 $\pm$ 0.03** | 10 |
| WET | **0.53 $\pm$ 0.05** | **0.57 $\pm$ 0.05** | **0.60 $\pm$ 0.07** | **0.35 $\pm$ 0.08** | 5 |
| Mean | **0.44 $\pm$ 0.03** | **0.51 $\pm$ 0.03** | **0.52 $\pm$ 0.03** | **0.28 $\pm$ 0.03** | 58 |
| *Monthly anomalies* | | | | | |
| ENF | **0.06 $\pm$ 0.02** | 0.05 $\pm$ 0.03 | **0.07 $\pm$ 0.02** | **0.02 $\pm$ 0.01** | 16 |
| EBF | 0.03 $\pm$ 0.02 | 0.04 $\pm$ 0.03 | 0.04 $\pm$ 0.02 | 0.07 $\pm$ 0.03 | 3 |
| DBF | **0.11 $\pm$ 0.04** | **0.10 $\pm$ 0.03** | **0.10 $\pm$ 0.03** | **0.08 $\pm$ 0.02** | 10 |
| MF | 0.04 $\pm$ 0.03 | 0.02 $\pm$ 0.01 | 0.06 $\pm$ 0.03 | 0.01 $\pm$ 0.01 | 3 |
| OSH | 0.02 $\pm$ NA | 0.08 $\pm$ NA | 0.15 $\pm$ NA | 0.02 $\pm$ NA | 2 |
| GRA | **0.12 $\pm$ 0.05** | **0.10 $\pm$ 0.04** | **0.12 $\pm$ 0.04** | **0.10 $\pm$ 0.01** | 9 |
| CRO | **0.10 $\pm$ 0.02** | **0.13 $\pm$ 0.02** | **0.15 $\pm$ 0.03** | 0.05 $\pm$ 0.03 | 10 |
| WET | **0.06 $\pm$ 0.02** | 0.03 $\pm$ 0.01 | 0.03 $\pm$ 0.01 | 0.05 $\pm$ 0.03 | 5 |
| Mean | **0.08 $\pm$ 0.01** | **0.08 $\pm$ 0.01** | **0.09 $\pm$ 0.01** | **0.05 $\pm$ 0.01** | 58 |

*3.2. Monitoring Spatial Variability, Seasonality, Interannual Variability, and Trends*

Mean annual GPP and average peak-season RSVIs for each site were not significantly correlated (Figure 4a). In contrast, average annual CCI and NDVI were positively correlated with average GPP for each site, albeit explaining only a small proportion of the spatial variability of GPP (Figure 4b). Our results instead indicated good relationships between the seasonality of the RSVIs and the seasonality of GPP (Figure 5). EVI and NIR$_v$ predicted the seasonality of GPP the best, accounting for almost half of the spatial variability of GPP seasonality. EVI, NIR$_v$, and NDVI had similar ranges of seasonality amongst the sites, but CCI had a much smaller range of seasonality.

**Table 2.** Summary of the statistical models indicating the controls of the performances of the RSVIs explaining weekly and monthly GPP and their standardised anomalies. Values indicate the standardised coefficients (β weights) ± standard error of the mean of each predictor except for the vegetation type (IGBP), indicating the vegetation type with the statistically best performance. GRA, grassland; MAT, mean annual temperature; VPD, mean annual vapour-pressure deficit; GPP, average GPP; GPP PV, seasonality of GPP; RSVI PV, seasonality of the vegetation indices; GPP PV%, variance explained by the seasonality of GPP; RSVI PV%, variance explained by the seasonality of the vegetation indices; GPP IPV, interannual variability of GPP; RSVI IPV, interannual variability of the vegetation indices.

| | MAT | VPD | IGBP | GPP | GPP PV | RSVI PV | GPP PV% | RSVI PV% | GPP IPV | RSVI IPV | $R^2$ |
|---|---|---|---|---|---|---|---|---|---|---|---|
| | | | | | | *Weekly* | | | | | |
| NDVI | | | GRA | | 0.49 ± 0.12 | | 0.45 ± 0.11 | 0.24 ± 0.08 | | | 0.81 |
| EVI | | | | | | | 0.63 ± 0.08 | 0.51 ± 0.07 | 0.26 ± 0.08 | | 0.76 |
| NIR$_v$ | | | | | | | 0.60 ± 0.09 | 0.47 ± 0.09 | 0.25 ± 0.09 | | 0.72 |
| CCI | | | | 0.36 ± 0.12 | 0.44 ± 0.12 | 0.47 ± 0.10 | | | | | 0.48 |
| | | | | | | *Weekly anomalies* | | | | | |
| NDVI | | | | | | | | | | | 0.00 |
| EVI | | 0.24 ± 0.12 | | | | | | | 0.27 ± 0.12 | 0.43 ± 0.13 | 0.33 |
| NIR$_v$ | | | | | | | | | 0.41 ± 0.12 | | 0.17 |
| CCI | | 0.50 ± 0.12 | | | | | | | 0.25 ± 0.12 | | 0.29 |
| | | | | | | *Monthly* | | | | | |
| NDVI | | | | | | | 0.95 ± 0.09 | | 0.56 ± 0.09 | | 0.68 |
| EVI | | | | | | | 0.63 ± 0.09 | 0.52 ± 0.09 | 0.19 ± 0.09 | | 0.65 |
| NIR$_v$ | | | | | | | 0.67 ± 0.12 | 0.31 ± 0.10 | 0.30 ± 0.11 | | 0.59 |
| CCI | | | | 0.38 ± 0.13 | | | 0.47 ± 0.12 | 0.50 ± 0.10 | 0.36 ± 0.15 | | 0.48 |
| | | | | | | *Monthly anomalies* | | | | | |
| NDVI | | | | | | | | | | | 0.00 |
| EVI | 0.34 ± 0.15 | | | | 0.43 ± 0.15 | | | | | 0.31 ± 0.15 | 0.21 |
| NIR$_v$ | | | | | | | | | 0.30 ± 0.13 | | 0.09 |
| CCI | | | | | 0.33 ± 0.14 | | −0.51 ± 0.14 | | | | 0.20 |

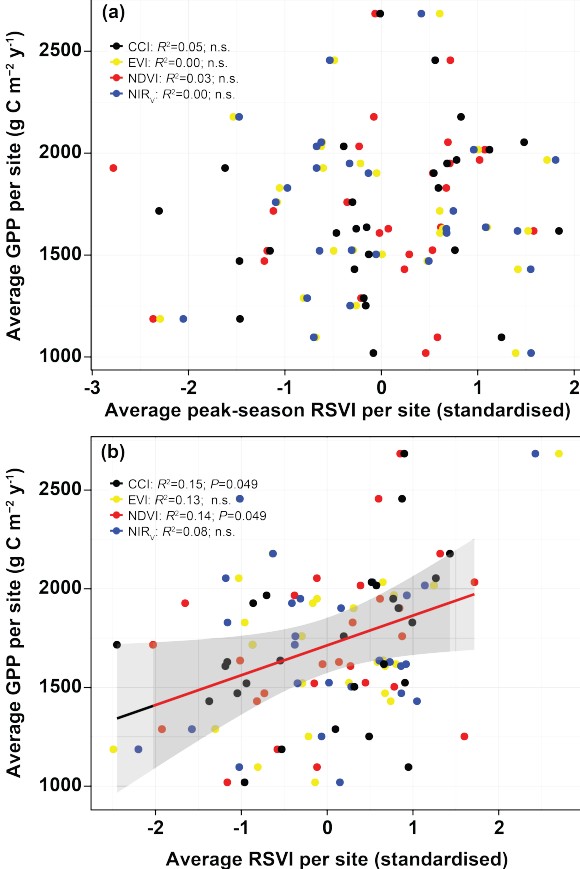

**Figure 4.** Spatial relationships between average GPP and the four remotely sensed vegetation indices (RSVI) using the average using peak-season (**a**) and yearly (**b**) averages. Shaded areas represent the standard error of the slopes. Nonsignificant relationships are indicated with "n.s.".

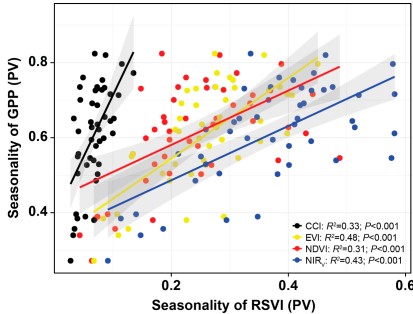

**Figure 5.** Spatial relationships between GPP seasonality and the seasonalities of the four remotely sensed vegetation indices (RSVI). Seasonality was calculated as the intra-annual PV of the monthly averages. Shaded areas represent the standard error of the slopes.

The average interannual variability of GPP for each site was not well monitored by the average interannual variability of the RSVIs: neither peak-season nor yearly average interannual variability of the RSVIs were significantly correlated with the interannual variability of GPP (Figure 6). Again, the range of variability was clearly smaller for CCI than the other RSVIs, indicating a lower interannual variability than for the other RSVIs. The trends of annual GPP were similarly not well monitored by any of the RSVIs, using either peak-season values or yearly averages (Figure 7). All indices had negative slopes with the trends in annual GPP when the trends of the yearly average RSVI values were used, being even significant for NDVI (Figure 7b).

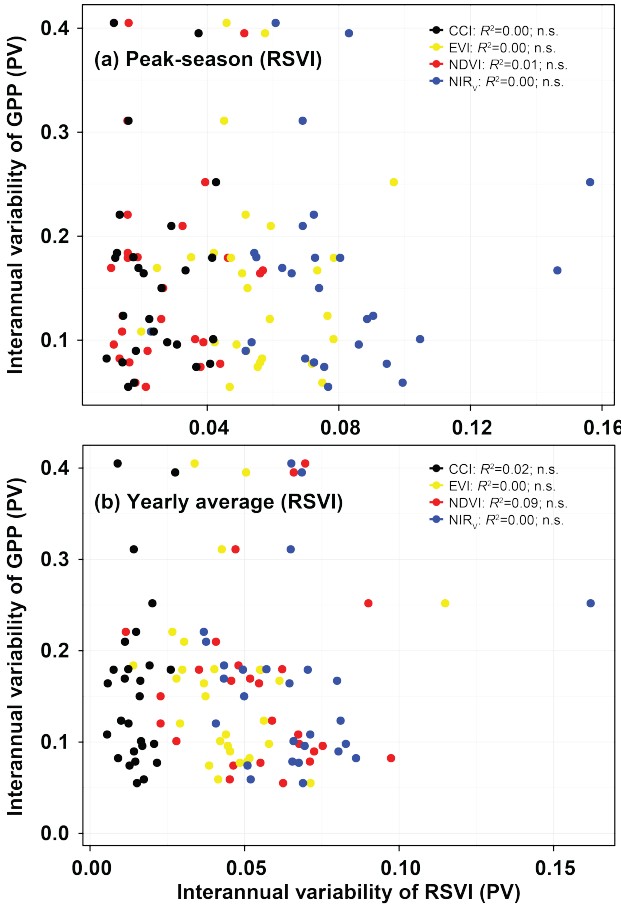

**Figure 6.** Spatial relationships between the interannual variability of GPP and the interannual variabilities of peak-season (**a**) and yearly average (**b**) remotely sensed vegetation indices. Nonsignificant relationships are indicated with "n.s.".

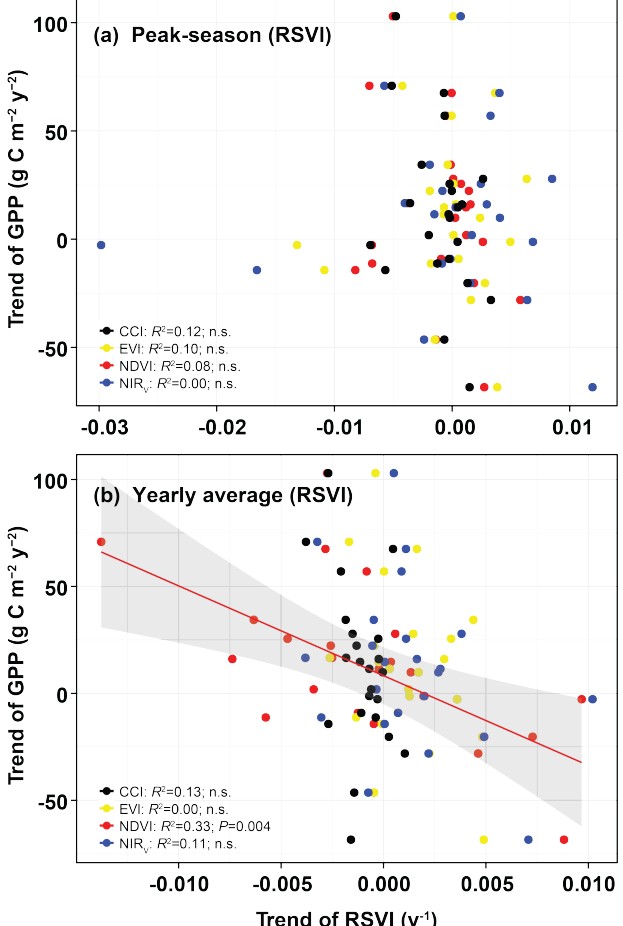

**Figure 7.** Spatial relationships between the trends of GPP and the trends of the peak-season (**a**) and yearly average (**b**) remotely sensed vegetation indices (RSVI). Only sites with seven or more years of data for gross primary production (GPP) and RSVI are used. Shaded areas represent the standard error of the slopes. Nonsignificant relationships are indicated with "n.s.".

## 4. Discussion

### 4.1. Seasonality Induces Temporal Correlations between GPP and the RSVIs

Our results indicated that all four RSVIs estimated GPP well when weekly and monthly data were used (Figure 3, Table 1). These findings are consistent with previous studies that have investigated the relationship between productivity and RSVIs [9,10,23,24]. We also found, however, a sharp decrease in the performances of the RSVIs when the seasonal component was removed from the time series: the correlations between the temporal anomalies of the RSVIs and GPP were very weak (Table 1). These results remained unaltered even when winter values were removed from the analyses (Table S2), perhaps because GPP and the RSVIs may all share a very similar seasonal component that is also correlated mostly with the amount of photosynthesis at the site level. This hypothesis was further supported by our findings indicating that the larger the variance explained by the seasonalities of GPP and the RSVIs, the better the temporal correlation between the RSVIs and GPP when the time series included the seasonal component. The combined errors of the two sources (i.e., measurements from eddy-covariance towers, about 30% of the annual amount [37], and the calculations of the RSVIs) were larger than the variability of the anomalies in the RSVIs and GPP, masking a good relationship between both sets of data. This finding was also supported by the better correlations between the temporal anomalies of the RSVIs and GPP at sites with a higher interannual variability of GPP, especially using weekly deseasonalised time series. Further improvements in both fields (i.e., eddy covariance and

remote sensing) will help to improve the performance of RSVIs when monitoring GPP. The capacity to infer anomalies of GPP from RSVIs using the MAIAC data set and our methodology is currently low, accounting only for ≤15% of their variability in the best cases (Table 1). The correlation between RSVIs and GPP may also be hindered because spectral signatures may react with a certain delay to changes in rainfall and temperature compared to the resulting GPP.

The temporal aggregation of the data, however, was not decisive for monitoring GPP. The weekly and monthly time series performed very similarly, albeit correlations were slightly stronger when monthly data were used. These results may be due to the smoothing of noise, artefacts, and outliers over larger sets of data when longer time periods are aggregated, decreasing the variance of the time series. Similarly, all four RSVIs provided very similar results, especially NDVI, EVI, and NIR$_V$. CCI, however, always had the lowest correlations for all ecosystem types (Table 1), even for evergreen needle-leaved forests, for which CCI has very accurately monitored phenology on a daily scale [9]. Using daily data in our analyses would likely have improved the performance of CCI, although its performance improved slightly when monthly instead of weekly data were used.

The performances of the RSVIs differed little amongst the ecosystem types. Evergreen broadleaved forests, open shrublands, and croplands had the lowest correlations with the RSVIs (Table 1). The RSVIs generally performed better for grasslands, even when deseasonalised time series were used. This result may indicate that these ecosystem types behaved differently than the other types. Grasslands often respond quicker to adverse environmental conditions (e.g., drought) than do evergreen broadleaved forests, which can sustain the same green biomass despite reductions in GPP due to adverse conditions. The higher spatial homogeneity of grassland ecosystems may also improve estimates of greenness. The two open shrublands in this study (ES-LgS and ES-Ln2, Table S1) are Mediterranean shrublands with very low vegetation densities and subjected to very high water deficits that reduce photosynthesis [38], even though the vegetation may remain green [12]. Bare soil can also potentially bias RSVI values [39]. The evergreen broadleaved forests were also Mediterranean (FR-Pue, IT-Cp2, IT-Cpz) with typically low RSVI seasonalities and water deficits during summer [40,41]. These typically Mediterranean characteristics may substantially reduce the performance of RSVIs in monitoring GPP in these types of ecosystems, in contrast to previous studies that found good correlations between productivity and RSVIs monitoring tree growth [12] and fruit production [2] in similar Mediterranean forests.

### 4.2. Performance of the RSVIs in Explaining the Spatial Variabilities of Seasonality, Interannual Variability, and Trends of GPP

Site-average NDVI and CCI monitored the spatial variability of site-average GPP moderately well, but EVI and NIR$_V$ did not (Figure 4). None of the RSVIs using average peak-season values for each site estimated productivity well. The difference in the performance when the average peak-season and annual average for each site were used may have been due to the larger effect of some circumstances on peak-season values over a short period of time, and annual averages may better integrate the overall annual state of the ecosystem. Based on our results, we would recommend using annual averages of NDVI and CCI to infer the productivities of different sites. The spatial variability of seasonality was monitored much better than average GPP (Figure 5). All indices performed well, but EVI and NIR$_V$ performed better than CCI and NDVI. These differences may be due to the higher sensitivity of EVI than NDVI to high levels of greenness, which would improve monitoring during the greenest periods of the growing season when NDVI can saturate. NIR$_V$ may then behave similarly to EVI [24]. In contrast, CCI may not perform as well as EVI and NIR$_V$ due to its small range of variability of seasonality amongst the sites.

Our results indicated that sites where the interannual variabilities of the RSVIs were higher did not have higher interannual variabilities of GPP (Figure 6). The range of interannual variabilities was also clearly lower for the RSVIs than GPP. The RSVI estimates were less variable amongst years than the estimates of GPP, which may account for the lack of a significant correlation. The trends of GPP

were also not well monitored by the RSVIs (Figure 7). Even the trends of NDVI were significantly negatively correlated with the trends of GPP (Figure 7b). Several studies have previously found that RSVIs were useful for monitoring the trends of the growing season, browning, or greening [42,43]. Our results, however, suggest that the trends of GPP may not follow the same trends of growing seasons or greening. This discrepancy may arise because: (i) equal green standing biomasses may photosynthesise differently depending on the environmental conditions, as discussed above (e.g., in Mediterranean sites experiencing droughts) or (ii) the combination of the two sources of error (GPP and RSVIs) were too large to correctly assess interannual trends, especially for single pixels. Using larger areas of land may have provided more positive results, similar to previous reports. Our results nonetheless contradict previous reports suggesting that RSVIs calculated using data from the MAIAC database improve their performance for the standard MODIS products [44]. Research comparing both sources of data is therefore warranted.

## 5. Conclusions

Overall, our results indicate that RSVI are a suitable tool to track different features of GPP. Grasslands were the ecosystems best predicted by RSVI, while evergreen broad-leaved forests were the worst. On average across different ecosystems, EVI and $NIR_v$ indices performed slightly better than NDVI, while CCI performance was clearly lower. However, when removing seasonality from the time series of GPP and RSVI, the correlation decreased considerably, suggesting that the main source of correlation between both was related to the seasonality of the site, as confirmed by our statistical models. We found that spatial variability in annual GPP was well captured by NDVI and CCI while seasonality was well captured by all RSVIs. However, spatial variability in trends and in interannual variability of GPP was not well captured by RSVI according to our analyses.

**Supplementary Materials:** The following are available online at http://www.mdpi.com/2072-4292/11/7/874/s1, Table S1: List of the sites with starting and ending dates (FLUXNET Tier 1). Vegetation types (IGBP); ENF, evergreen needleleaved forest; EBF, evergreen broadleaved forest; DBF, deciduous broadleaved forest; MF, mixed forest; OSH, open shrubland; GRA, grassland; CRO, cropland; WET, permanent wetland. Table S2: Amounts of variance explained ($R^2$, mean $\pm$ standard error of the mean) for the raw and deseasonalised weekly and monthly time series of GPP by NDVI, EVI, $NIR_v$, and CCI after removing winter weeks (1–12 and 46–52) and months (December, January, February, and March). Significant coefficients are highlighted in bold. ENF, evergreen needleleaved forest; EBF, evergreen broadleaved forest; DBF, deciduous broadleaved forest; MF, mixed forest; OSH, open shrubland; GRA, grassland; CRO, cropland; WET, permanent wetland. "NA" stands for not available.

**Author Contributions:** M.F.-M., R.Y., J.G., and J.P. conceived and design the paper. M.F.-M. analysed the data. All authors contributed substantially to the writing of the manuscript.

**Funding:** This research was funded by the Spanish Government project CGL2016-79835-P (FERTWARM), the European Research Council Synergy grant ERC-2013-726 SyG-610028 IMBALANCE-P, and the Catalan Government project SGR 2017-1005. M.F.-M. is a postdoctoral fellow of the Research Foundation-Flanders (FWO). M.B. acknowledges the support provided by the EU Horizon 2020 Research and Innovation programme under the Marie Skłodowska-Curie grant (INDRO, grant no. 702717). Support from the NASA ABoVE program (award #NNX15AT78A) for JG, RY, and GH is also acknowledged. The APC was funded by M.F-M's FWO postdoctoral fellowship.

**Acknowledgments:** This work used eddy-covariance data acquired and shared by the FLUXNET community, including the networks AmeriFlux, AfriFlux, AsiaFlux, CarboAfrica, CarboEuropeIP, CarboItaly, CarboMont, ChinaFlux, Fluxnet-Canada, GreenGrass, ICOS, KoFlux, LBA, NECC, OzFlux-TERN, TCOS-Siberia, and USCCC. The data from the ERA-Interim reanalysis were provided by ECMWF and processed by LSCE. The FLUXNET eddy-covariance data were processed and harmonised by the European Fluxes Database Cluster, AmeriFlux Management Project, and Fluxdata project of FLUXNET, with the support of CDIAC, the ICOS Ecosystem Thematic Center, and the offices of OzFlux, ChinaFlux, and AsiaFlux.

**Conflicts of Interest:** The authors declare no conflict of interest.

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
