# Peer review of "Monitoring Spatial and Temporal Variabilities of Gross Primary Production Using MAIAC MODIS Data"

_remotesensing, doi:10.3390/rs11070874_

Round 1
Reviewer 1 Report
This is a very interesting paper that evaluated the ability of remotely sensed vegetation indices to monitor different faces of terrestrial productivity. The main contribution resides in the assessment of multiple indices, multiple aspects of GPP, and across multiple ecosystem types. There are a lot of topics covered here, and I congratulate the authors for such a big effort.
It is difficult for the reader, however, to retain the key findings of this study after reading the paper. Part of the reason is that this is an ambitious paper, and as such there is a lot of information and analyses presented. The discussion tries to cover all aspects, but fails at summarizing the main findings and contributions in a condensed way. My main suggestion, therefore, is to expand the discussion with a final table, or figure, or bullet points summarizing the key findings, lessons, and recommendations from this study, and trying to highlight what agrees or disagree with previous knowledge and what is new. I believe such piece with be valuable contribution from this paper, and useful both for remote sensing and ecologists.
Finally, the abstract and discussion sections would benefit from stronger, finishing sentences. The abstract currently finishes with results, and the discussion finishes with caveats and recommendations of future work. Stronger sentences would be helpful there.
Author Response
Reviewer 1
Q1: This is a very interesting paper that evaluated the ability of remotely sensed vegetation indices to monitor different faces of terrestrial productivity. The main contribution resides in the assessment of multiple indices, multiple aspects of GPP, and across multiple ecosystem types. There are a lot of topics covered here, and I congratulate the authors for such a big effort.
R1: We thank the reviewer for such a positive evaluation of our work and his/her time spent reading it.
Q2: It is difficult for the reader, however, to retain the key findings of this study after reading the paper. Part of the reason is that this is an ambitious paper, and as such there is a lot of information and analyses presented. The discussion tries to cover all aspects, but fails at summarizing the main findings and contributions in a condensed way. My main suggestion, therefore, is to expand the discussion with a final table, or figure, or bullet points summarizing the key findings, lessons, and recommendations from this study, and trying to highlight what agrees or disagree with previous knowledge and what is new. I believe such piece with be valuable contribution from this paper, and useful both for remote sensing and ecologists.
R2: Following the reviewer’s advice, we have now added a conclusion section at the end of the manuscript (L.369-378), summarizing the main findings of the paper: “Overall, our results indicate that RSVIs are a suitable tool to track different features of GPP. Grasslands were the ecosystems best predicted by RSVI, while evergreen broad-leaved forests were the worst. On average across different ecosystems, EVI and NIRv indices performed slightly better than NDVI, while CCI performance was clearly lower. However, when removing seasonality from the time series of GPP and RSVI, the correlation decreased considerably, suggesting that the main source of correlation between both was related to the seasonality of the site, as confirmed by our statistical models. We found that spatial variability in annual GPP was well captured by NDVI and CCI while seasonality was well captured by all RSVIs. However, spatial variability in trends and in interannual variability of GPP was not well captured by RSVI according to our analyses”.
Q3: Finally, the abstract and discussion sections would benefit from stronger, finishing sentences. The abstract currently finishes with results, and the discussion finishes with caveats and recommendations of future work. Stronger sentences would be helpful there.
R3: We have now rephrased the end of the abstract with stronger messages, L.36-38: “Overall, our results indicate that RSVIs are suitable to track different facets of GPP variability at the local scale, therefore being reliable sources to monitor GPP features at larger geographical scales”. We now also end our manuscript with a conclusions section (see reply above).
Reviewer 2 Report
I read with strong interest the paper. The possibility to correlate Vegetation Indices with biophysical ad physical parameters is fascinating, since remote sensing can help in mapping the spatio-temporal variation of these variables. The paper is very well written and organised. In my opinion the authors should better stress and explain the need to remove the seasonality: why do this ? what are you looking for? What are the meaning of the anomalies you are lookong for? Are you trying to assess trends ? Are you looking for trends in time? I would suggest to introduce some sentences to help the reader to understand.
Furthermore, I think that vegetation canopies modify their spectral signatures with a certain lag time compared with rainfall, temperature, etc. signals. This would mean that some correlation does not work well because the Vegetation indices signal is shifted in time compared with other signals. In my opinion authors should also write a short paragraph about.
There is an appendix (Supplementary material) that can be useful, but I suggest to move the Figure S1 in the main part of the manuscript. The following table S1 could be maintained in the Appendix but I think the references of this Appendix should have a different numbering systems that the main paper.
Author Response
Q1: I read with strong interest the paper. The possibility to correlate Vegetation Indices with biophysical ad physical parameters is fascinating, since remote sensing can help in mapping the spatio-temporal variation of these variables. The paper is very well written and organised. In my opinion the authors should better stress and explain the need to remove the seasonality: why do this ? what are you looking for? What are the meaning of the anomalies you are lookong for? Are you trying to assess trends? Are you looking for trends in time? I would suggest to introduce some sentences to help the reader to understand.
R1: We thank the reviewer for such a positive evaluation of our work and his/her time spent reading it. We now explain the need to remove seasonality, why we are doing so and what is the meaning of the anomalies in L.172-177: “Our results would suggest that the RSVIs were able to track the temporal variability of GPP independently of their seasonal cycle if the models using the standardised anomalies performed as well as those using data with the seasonal component. If the performance using anomalies was dramatically lower than the performance using data with the seasonal component, our results would identify the RSVIs that could track the seasonal component of GPP well, but not its seasonal anomalies (deviations from the mean)”. Trends were assessed using a different analysis, see L.213-217: “Finally, we determined if the trends in annual GPP could be inferred from the trends in the annual average and the annual peak-season values of the RSVIs by calculating the annual trends of GPP and the annual average and peak-season RSVIs of the sites for which the time series were at least seven years [33]. Trends were calculated using the robust Theil-Sen’s slope estimator [34]”.
Q2: Furthermore, I think that vegetation canopies modify their spectral signatures with a certain lag time compared with rainfall, temperature, etc. signals. This would mean that some correlation does not work well because the Vegetation indices signal is shifted in time compared with other signals. In my opinion authors should also write a short paragraph about.
R2: Following the reviewer’s advice, we have included some discussion about the fact that vegetation canopy can react to changes in rainfall, temperature etc.. with a certain delay, see L.282-284: “The correlation between RSVIs and GPP may also be hindered because spectral signatures may react with a certain delay to changes in rainfall and temperature compared to the resulting GPP”.
Q3: There is an appendix (Supplementary material) that can be useful, but I suggest to move the Figure S1 in the main part of the manuscript. The following table S1 could be maintained in the Appendix but I think the references of this Appendix should have a different numbering systems that the main paper.
R3: We have now moved Figure S1 into the main paper (now Figure 1) and changed the numbering system of the references in the supplementary material as suggested by the referee.
Reviewer 3 Report
The paper aims at comparing the temporal patterns of four vegetation indices (NDVI, EVI, NIRv, and CCI) calculated on weekly, monthly, and annual bases and GPP derived from 58 eddy-flux towers distributed throughout Europe. The intent was challenging and the authors managed to handle it satisfactorily both from a scientific and stylistic point of view. The paper faces a very interesting issue that greatly contributes to the understanding of the ecological meaning and reliability of the most used vegetation indices. I do believe that the paper is suitable for publication after some moderate revisions.
In general (please, check the pdf attached for details):
- I do believe that a flowchart of the methodological steps would greatly support the comprehension of the logical flow of all the analysis and prevent confusion. It is hard to follow all the steps without a schematic reference. Please, add it.
- How did the authors manage the spatial mismatch between the fluxes (point) data and the 1-km pixel data? How large is the flux footprint covered by the towers? Please, explain it clearly because it is a key pint for the reliability of the study.
- Why did the authors use the average for the daily values, while then they use the median and not the mean for the weekly and monthly values? Why they did not use the maximum, for instance? Please, explain the rationale behind these choices.
-L175-190: How did the authors chose such thresholds of years (5 and 7)?
- What about the climatic data? The authors used them in the statistical analysis but they are not mentioned in the Results nor the in the Discussion sections.
- L297: May the authors justify this contrasting evidences?
- A paragraph about the implications of the evidences derived from this study is missing but required in order to highlight its applicability and usefulness.

Author Response

(The authors gave the same response as above.)
